# Multimodal Treatments for Brain Metastases from Renal Cell Carcinoma: Results of a Multicentric Retrospective Study

**DOI:** 10.3390/cancers15051393

**Published:** 2023-02-22

**Authors:** Pierina Navarria, Federico Pessina, Giuseppe Minniti, Ciro Franzese, Beatrice Marini, Giuseppe D’agostino, Marco Badalamenti, Luca Raspagliesi, Giacomo Reggiori, Francesca Lobefalo, Laura Fariselli, Davide Franceschini, Luisa Bellu, Elena Clerici, Valentina Pinzi, Marta Scorsetti

**Affiliations:** 1Radiotherapy and Radiosurgery Department, IRCCS Humanitas Research Hospital, Rozzano, 20089 Milan, Italy; 2Neurosurgery Department, IRCCS Humanitas Research Hospital, Rozzano, 20089 Milan, Italy; 3Department of Biomedical Sciences, Humanitas University, 20089 Milan, Italy; 4Department of Radiological Sciences, Oncology & Pathology, Sapienza University of Rome, 20089 Rome, Italy; 5Radiotherapy Unit, Istituto Neurologico Fondazione “Carlo Besta”, 20089 Milan, Italy

**Keywords:** brain metastases, renal cell carcinoma, radiosurgery, hypofractionated radiosurgery, surgery

## Abstract

**Simple Summary:**

Around 2–15% of primary renal cell carcinoma patients (RCC) will develop brain metastases (BMs) during the disease course. The prognosis of brain metastatic RCC patients is poor when compared with extracranial metastatic patients, and determining the optimal local therapeutic approach is a challenge. The aim of our retrospective multi-institutional study was to assess the efficacy of local treatments, radiosurgery in single or multiple fractions (SRS/HSRS) with or without surgery, and to identify prognostic factors eventually conditioning outcome. Patients with limited BMs (up to four) were treated with single-dose SRS in cases of small lesions, HSRS for large BMs unsuitable for surgical resection, or surgical resection followed by SRS/HSRS. We confirmed the efficacy and safety of SRS/HSRS in 120 patients analyzed for 136 BMs treated. Patients with favorable/intermediate International Metastatic Database Consortium (IMDC) score, with a higher RCC-graded prognostic assessment (GPA) score, with an early occurrence of BMs from primary diagnosis, with absence of extracranial metastases, and who underwent a combined local treatment (surgery plus adjuvant HSRS) had a better outcome.

**Abstract:**

The aim of this study was to evaluate the clinical outcomes of a large series of brain metastatic renal cell carcinoma (BMRCC) patients treated in three Italian centers. Methods: A total of 120 BMRCC patients with a total of 176 lesions treated were evaluated. Patients received surgery plus postoperative HSRS, single-fraction SRS, or hypofractionated SRS (HSRS). Local control (LC), brain distant failure (BDF), overall survival (OS), toxicities, and prognostic factors were assessed. Results: The median follow-up time was 77 months (range 16–235 months). Surgery plus HSRS was performed in 23 (19.2%) cases, along with SRS in 82 (68.3%) and HSRS in 15 (12.5%). Seventy-seven (64.2%) patients received systemic therapy. The main total dose and fractionation used were 20–24 Gy in single fraction or 32–30 Gy in 4–5 daily fractions. Median LC time and 6 month and 1, 2 and 3 year LC rates were nr, 100%, 95.7% ± 1.8%, 93.4% ± 2.4%, and 93.4% ± 2.4%. Median BDF time and 6 month and 1, 2 and 3 year BDF rates were n.r., 11.9% ± 3.1%, 25.1% ± 4.5%, 38.7% ± 5.5%, and 44.4% ± 6.3%, respectively. Median OS time and 6 month and 1, 2 and 3 year OS rates were 16 months (95% CI: 12–22), 80% ± 3.6%, 58.3% ± 4.5%, 30.9% ± 4.3%, and 16.9% ± 3.6, respectively. No severe neurological toxicities occurred. Patients with a favorable/intermediate IMDC score, a higher RCC-GPA score, an early occurrence of BMs from primary diagnosis, absence of EC metastases, and a combined local treatment (surgery plus adjuvant HSRS) had a better outcome. Conclusions: SRS/HSRS is proven to be an effective local treatment for BMRCC. A careful evaluation of prognostic factors is a valid step to manage the optimal therapeutic strategy for BMRCC patients.

## 1. Introduction

The increasing incidence of renal cell carcinoma (RCC) and the prolonged survival associated with development of more effective systemic therapies have led to a greater observation of metastatic spread, most commonly into lung, liver, or bone [1,2]. Brain metastasis (BM) is a more infrequent event, ranging from 2% to 15%, occurring late during the disease course, and usually accompanied by a diffuse extracranial metastatic involvement [3,4]. Notwithstanding the metastatic disease status, the outcome of extracranial metastatic RCC patients remains satisfactory with about 50% survival after 5 years, unlike the brain metastastic disease patients, characterized by a worse prognosis with an median overall survival (OS) time that does not exceed 10 months [5,6,7,8,9,10,11]. The presence of BMs represents an exclusion criterion in a large part of randomized prospective trials due to the poor local control (LC) in employing systemic treatment and the poor outcome of these patients [12]. Indeed, most of the available data come from retrospective observation, reflecting the difficulty in drawing any conclusions concerning the management of these patients. Currently, brain metastatic RCC (BMRCC) patients should receive brain-directed local therapy with radiation therapy and/or surgery, as suggested by international guidelines, considering the limited intracranial efficacy of the current systemic therapy agents [13]. Surgical resection followed by adjuvant radiosurgery in single (SRS) or multiple fractions (HSRS) on the tumor bed is indicated in selected patients with single and large BMs, a good KPS, and limited systemic disease. Results of randomized trials showed a local control in site of treatment greater than 80% with improvement in neurocognitive functions [14,15,16,17]. For patients with small and/or multiple BMs, or who are unsuitable for surgery, SRS or HSRS is able to provide excellent local control with negligible toxicity [18,19,20,21,22,23,24,25]. Unfortunately, in studies leading to the current standard of care for brain metastatic patients, a small number (4–17%) of the included patients had a primary diagnosis of RCC, and, to date, there has been no prospective study analyzing the outcome of this subgroup. However, considering the availability of more effective systemic therapy able to obtain a better extracranial disease control, as well as the most common application of brain MRI to detect small brain lesions, an increase in BMRCC patients is expected in the next few years. Given the paucity of data in this setting, this is an important area for further research, and treating patients with mRCC harboring BMs remains a challenge. On the basis of this background, we evaluated the outcome of a large series of BMRCC patients treated in three Italian Institutions aiming to evaluate efficacy of local treatments (SRS/HSRS with or without surgery), and to identify prognostic factors eventually conditioning outcome. 

## 2. Material and Methods 

### 2.1. Patients and Procedures

The present study includes patients with limited BMs (up to four) from RCC treated with single-dose SRS or HSRS (3–5 fractions) in relation to the number and cumulative tumor volume of the brain lesions. Single-dose SRS was employed in cases of small lesions (≤3 cm) or multiple BMs with a cumulative tumor volume up to 15 cm^3^; HSRS was administered for large BMs (>3 cm) unsuitable for surgical resection in relation to patient age, performance status, and diffuse and uncontrolled extracranial metastases. Surgical resection followed by SRS/HSRS was performed in select cases characterized by KPS 90–100, controlled extracranial disease, single BM with maximum diameter ≥21 mm, presence of two BMs in which one was larger and conditioning a mass effect, life expectancy longer than 3 months, and progressive neurological deficits or seizures. All patients were treated in agreement with the Helsinki declaration. This study was based on a retrospective analysis of treatment charts and received approval by the local Ethics Committee. 

### 2.2. Treatment: SRS/HSRS

For target volume delineation, contrast-enhanced T1MRI and CT scans were acquired, and images were co-registered. For patient immobilization and repositioning, a frameless system was employed. The gross target volume (GTV) corresponded to the BM volume; the planning target volume (PTV) was generated by adding isotropic margins from GTV of 0–3 mm. The delineated organs at risk (OARs) were the brain, brainstem, optic nerves, chiasm, and lenses. The total doses and fractionation planned according to the maximum diameter of BMs were 24 Gy/1 fraction for lesions up to 2 cm, 20 Gy/1 fraction for lesions up to 2–3 cm, 32 Gy/4 fractions for BMs 3.1–5 cm, and 30 Gy/5 fraction for larger ones; following surgical resection, a dose of 30 Gy in three fractions on the tumor bed was administered. Patients were treated with the volumetric modulated arc technique RapidArc (LINAC or Cyber Knife); Exactrac (Brainlab) and/or cone beam CT imaging was performed daily for patient setup and positioning verification. 

### 2.3. Systemic Therapy

Some patients received systemic adjuvant therapy after local treatments of BMs. Different regimens were used in relation to previous treatments received, consisting of vascular endothelial growth factor receptor tyrosine kinase inhibitors (VEGFR-TKIs) or immune checkpoint inhibitors (ICIs) according to national and international guidelines. 

### 2.4. Outcome Evaluation

Clinical outcome was evaluated by neurological examination and brain MRI performed 2 months after RT and every 3 months thereafter. Local progression was defined as a radiographic increase in enhancing abnormalities in the irradiated volume on serial MR imaging according to RANO criteria, and BDF was defined as the presence of new BMs or leptomeningeal enhancement outside the irradiated volume [26]. Toxicities were assessed during treatment at 2 months after RT and every 3 months thereafter, and then graded according to Common Terminology Criteria for Adverse Events version 4.0. Systemic disease was evaluated by contrast-enhanced total body CT scan and 18-FDG CT-PET. 

### 2.5. Statistical Analysis

Standard descriptive statistics were used to describe the general data behavior. Survival and recurrence time observations were computed according to the method of Kaplan and Meier, starting from the date of BM treatments. In order to investigate the prognostic role of different individual variables, the log rank test and univariate Cox regression were used, respectively, for categorical and numerical variables age, gender, KPS, stage at diagnosis, grade of primary tumor, interval time between primary tumor diagnoses and appearance of brain metastases, RCC-specific GPA score, IMDC score, presence of other metastatic site at time of BMs, number, site and size of BMs, type of local treatment performed, and systemic treatment administration. The multivariate Cox model was used as a method to estimate the independent association of a variable set with local control (LC), brain distant progression (BDF), progression-free survival (PFS), and overall survival (OS). Statistical analysis was performed using Medcalc software, v17.7 (MedCalc software, Ostend, Belgium).

## 3. Results

### 3.1. Patients and Treatments

From April 2003 to July 2021, among 136 BMRCC patients treated, 120 were evaluable and, therefore, included in the present analysis. Eighty-eight (73.3%) were male and 32 (26.7%) were female. Patient, tumor, and treatment characteristics at diagnosis are shown in Table 1. 

Upon BM occurrence, a large number of patients had a good KPS (73.3%), presence of extracranial metastases (89.2%), an RCC-specific GPA (graded prognostic assessment) score 3.5–4 (47.5%), and an intermediate International Metastatic Database Consortium (IMDC) score (73.3%). The median interval time (IT) between the diagnosis of primary RCC and the appearance of BMs was 28 months (range 0–253 months); the total number of BMs irradiated was 176, and most patients had 1–2 BMs (91.7%). The treatments performed were surgical resection followed by SRS/HSRS on the tumor bed in 23 (19.2%) cases, SRS alone in 82 (68.3%), and HSRS only in 15 (12.5%). Seventy-seven (64.2%) patients received systemic therapy after BM treatment consisting of vascular endothelial growth factor receptor tyrosine kinase inhibitors (VEGFR-TKIs) and/or immune checkpoint inhibitors (ICIs). The most frequent drugs employed were pazopanib, cabozantinib, sunitinib nivolumab, and ibilimumab. Details are shown in Table 2.

Local control (LC), distant brain failure (BDF), progression-free survival (PFS), and overall survival (OS) analysis 

The median follow-up time from BM treatment was 77 months (range 16–235 months). Seven (5.8%) patients had local recurrence at the site of treatment at a median time of 9 months (range 7–16 months). The median LC time and 6 month and 1, 2, and 3 year LC rates were nr, 100%, 95.7% ± 1.8%, 93.4% ± 2.4%, and 93.4% ± 2.4%, respectively as shown in Figure 1.

Brain distant failure (BDF) occurred in 35 (29.2%) patients at a median time of 10 months (range 1–113 months). The median BDF time and 6 month and 1, 2, and 3 year BDF rates were n.r., 11.9% ± 3.1%, 25.1% ± 4.5%, 38.7% ± 5.5%, and 44.4% ± 6.3%, respectively. The median PFS time and 6 month and 1, 2, and 3 year PFS rates were 5 months (95% CI: 4–8), 50% ± 4.5%, 30% ± 4.1%, 9.9% ± 2.8%, and 4.1% ± 2.1%, respectively. The median OS time and 6 month and 1, 2, and 3 year OS rates were 16 months (95% CI: 12–22), 80% ± 3.6%, 58.3% ± 4.5%, 30.9% ± 4.3%, and 16.9% ± 3.6%, respectively. At the last observation time, 17 (14.2%) patients were alive, and 103 (85.8%) were dead, three (2.9%) from intracranial progression, 90 (87.4%) from extracranial progression, eight (7.8%) from extracranial and brain progression, and two (1.9%) from a tumor-unrelated cause. Figure 2 shows the BDF, PFS, and OS for all patients treated.

### 3.2. Prognostic Factors Analysis

Prognostic factors conditioning LC, BDF, PFS, and OS were analyzed. Gender, age, and stage at diagnosis were not recorded as the conditioning outcomes. LC was significantly influenced by the type of local treatment performed; in particular, patients receiving surgical resection followed by adjuvant HSRS had 6 month and 1, 2, and 3 year LC rates at treatment site of 100%, 100%, 96.2% ± 6.7%, and 96.2% ± 6.7 compared to 100%, 97.2% ± 1.9%, 95.2% ± 2.7%, and 95.2% ± 2.7% for SRS alone, and 100%, 80.8% ± 10.3%, 80.8% ± 10.3%, and 80.8% ± 10.3% for HSRS alone, respectively (p = 0.0258). No statistically significant factors were identified as impacting BDF. Patients harboring a favorable/intermediate IMDC score, a higher RCC-GPA score, an early occurrence of BMs from primary diagnosis, and an absence of EC metastases, who received a combined local treatment (surgery plus adjuvant HSRS), had a better outcome. Among these patients, the administration of systemic treatment, particularly ICIs, influenced survival. Details about prognostic factors statistically relevant for PFS and OS are shown in Table 3.

### 3.3. Salvage Treatment for Intracranial/Local Progression

Among seven local relapse patients, all received further treatments, consisting of surgery followed by adjuvant HSRS in two cases, HSRS in four cases, and SRS in one case. In almost all patients with BDP (32/35), a new local treatment was performed: 26 SRS, two HSRS, and four WBRT.

### 3.4. Toxicity

Treatment-related toxicities occurred in 23 (19.2%) patients and consisted of nausea, vomit, and headache in six (5%), partial or generalized seizure in seven (6%), motor deficit in four (3.3%), and grade 2–3 radionecrosis in six (5%) patients.

## 4. Discussion

The results of a retrospective multi-institution study assessing the role of local treatments in brain metastatic renal cell carcinoma (BMRCC) patients were provided. A total of 120 patients with 176 BMs treated were evaluated. SRS or HSRS alone was performed in 80% of patients, while 20% received surgical resection followed by SRS/HSRS on the tumor bed. Although RCC is considered a radioresistent tumor, the administration of SRS or HSRS, which enables the delivery of ablative RT doses in one or few fractions, led to satisfactory local control. Indeed, the 1 and 3 year LC rates were 96% and 93%, respectively, with only 7/176 (~4%) cases of relapse at the site of treatment. Our results are comparable to or better than published series [18,19,20,21,22,23,24]. Wardak et al. analyzed 38 patients treated with radiosurgery on 243 brain metastases [25]. The reported 1 and 2 year local control rates were 92% and 86%, respectively. Similarly, Klausner et al. treated 362 consecutive patients with single-dose SRS on brain metastases, using Gammaknife and Linac-based SRS mode [27]. The 1 and 3 year LC rates were 94% and 92%, respectively; significant predictive factors were a minimal dose >17 Gy and concomitant TKI treatment, administered in about one-third of the patients. In our series, single-dose SRS using a median dose of 24 Gy proved to be more effective compared to the hypofractionated schedule employing a median dose of 30 Gy in five fractions, although these data may have been affected by the different volume of BMs treated (larger in the HSRS). A more beneficial effect on LC was recorded using a combined treatment (surgery plus HSRS on the tumor bed), obtaining a durable LC in almost all patients treated (96%). Nonetheless, accurate patient selection is essential to plan the optimal local approach, particularly when surgical resection is being considered. Several scores exist to predict survival [8,9,10,11]. They involve some clinical features (KPS, age, extracranial metastatic spread, and interval time between initial RCC diagnosis and BM occurrence) and some BM characteristics (number of BMs, and cumulative intracranial tumor volume) associated with outcome. Patients with a single or limited BMs, controlled or limited systemic cancer, and active EC metastases suitable for local treatments would seem to be adequate candidates for a combined local treatment consisting of surgery followed by radiation therapy. In addition, although the patient’s neurological status can represent a critical parameter for survival, surgical resection can lead to relief of neurological symptoms by improving patient QoL. In our series, surgery was performed only in patients with single BM, good KPS, and absence or limited/controlled EC metastases. However, considering the results obtained, a wider patient selection criteria could be advisable, including patients with limited BMs (up to four) in which SRS might be administered on the other lesions before surgery, as well as patients suitable for surgery or SBRT on the EC metastatic site, or patients in which a systemic therapy could be performed, aiming to optimally control the whole disease. Survival was also highly satisfactory, with 1, 2, and 3 year OS rates of 58%, 31%, and 17%, respectively. In other published papers, the occurrence of BMs compared to EC was recorded as negatively influencing survival. Zaorsky et al., in their meta-analysis, showed a 1 year survival rate of 87% for extracranial metastatic patients, in contrast to 50% in cases of intracranial disease [28]. This better outcome could be related to a greater benefit observed when employing systemic therapy, which is less effective to control brain disease. Published series, assessing the role of systemic therapy rather than local treatment in BMRCC patients, failed to prove a clear beneficial effect on outcome. A phase II trial investigated the efficacy of sunitinib in 16 BMRCC patients ineligible for local treatment and naïve to prior systemic therapy, showing no beneficial effect on objective response, median time to progression, and median OS [29]. Similarly, data available using sunitinib and pazopanib, sorafenib, or axitinib have not proven any efficacy in brain disease control or survival [30,31,32,33]. More promising results have been recorded using cabozantinib, a TKI able to easily penetrate the BBB, as documented by preclinical models, retrospective analysis, and the preliminary results of the ongoing prospective phase II CABRAMET study (NCT03967522) [34,35,36,37]. The benefits and safety of ICIs have also been investigated [38,39,40,41,42]. The interim results of the ongoing CheckMate 920 study, aiming to assess the role of nivolumab plus ipilimumab in BMRCC patients, showed an ORR of 28.6%, and a median PFS of 9.0 months. The small sample size and the short follow-up time do not allow clearly concluding to the benefit of this treatment, but the preliminary results seem encouraging. The challenge is now to assess the optimal therapeutic approach, able to control both cranial and EC metastatic disease. The combination of TKIs and local therapy has the advantage of allowing a better control of systemic disease, and the possibility of carrying out the systemic treatment before the occurrence of BMs. On the other hand, the combination of RT and ICIs, through a complex immunologic synergic action, might also potentially improve the antitumoral response [43,44]. Chen et al., in a retrospective trial, assessed the outcome of 260 brain metastatic patients with a broad spectrum of malignancies (including 33 BMRCCs) [45]. Patients were treated with HFSRT/SRS alone, concurrent HFSRT/SRS + ICIs, or nonconcurrent HFSRT/SRS + ICIs. A significant improvement in OS and BDF was observed for concurrent HFSRT/SRS + ICIs compared to other groups, but no advantage in LC was determined. In our analysis, the administration of systemic therapy after local treatment (SRS/HSRS or surgery plus HSRS) showed a favorable effect on survival, with 3 year OS rates of 9% without treatment, 18% with TKIs, and 37% using ICIs (*p* = 0.0540). Our results confirm the need for a multimodal treatment approach. Indeed, patients receiving surgery plus SRS/HSRS and/or systemic therapy had better outcomes with more than one-third of patients alive at 3 years. In addition, patients without EC metastases at BM occurrence had a major advantage in terms of survival, reaching 44% at 3 years, suggesting that the control of all metastatic sites might improve the outcome of these patients with a poor prognosis. Unlike what is documented in the literature in our population, patients with an IT from primary diagnosis to occurrence of BMs shorter than 1 year had the better outcome. These data have to be interpreted with caution, and they appear difficult to explain. This finding might be related to a low disease burden in this group of patients and/or an early administration of systemic therapy. We are aware that our analyses have all the limitations of a retrospective study, including selected patients characterized by limited brain disease (up to four BMs), mainly 1–2 BMs, limited EC metastases, good KPS (90–100), intermediate IDMC score, and good RCC-GPA score, whereby surgery was possible in about 20% of cases, along with systemic therapy, TKIs or ICIs, administered in more than 60% of cases after local treatment. Notwithstanding, in our series, the main selection criterion for surgical resection was related to BM characteristics and extracranial disease status, and not exclusively to patients’ general conditions. In spite of this, to our knowledge, this is one of the larger series published in the literature evaluating the role of local treatments in BMRCC patients, and the results obtained are highly satisfactory, paving the way for new prospective approaches in patients, until now, considered to have a poor prognosis.

## 5. Conclusions

The presence of BMs in primary RCC patients define a metastatic disease with poor outcome. The control of brain disease is fundamental in allowing an acceptable QoL, to proceed with systemic therapy, potentially impacting survival. Our results underlined that SRS/HSRS is a safe and feasible treatment with satisfactory local control and negligible toxicity. A multimodal treatment approach including surgery, radiosurgery, and systemic therapy provided the best outcome. Further prospective studies are needed to identify patients in which a multimodal treatment approach can be considered.

## Figures and Tables

**Figure 1 cancers-15-01393-f001:**
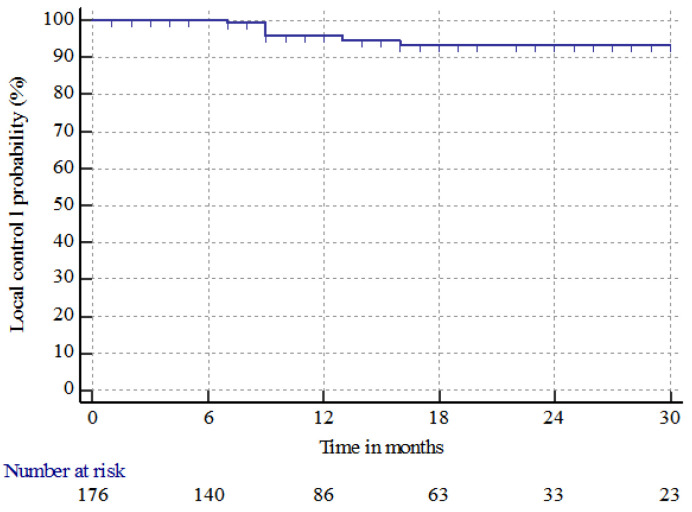
Local control (LC) of brain metastases in patients with renal cell carcinoma treated with radiosurgery in single or multifraction (SRS/HSRS) or surgery followed by HSRS.

**Figure 2 cancers-15-01393-f002:**
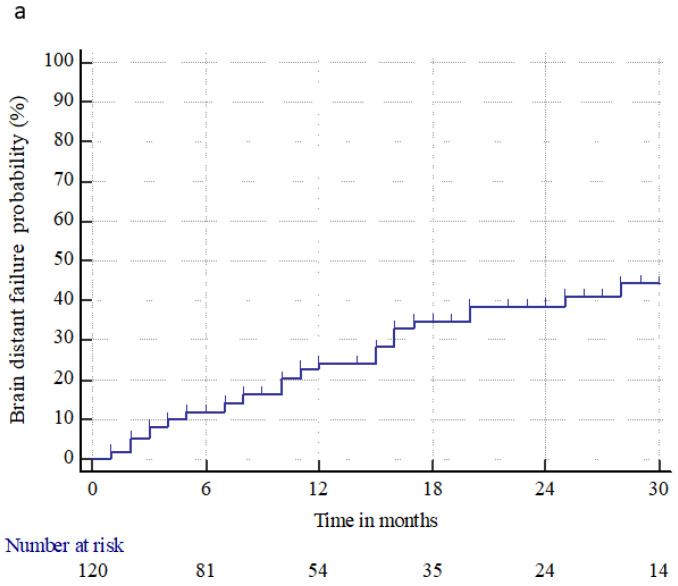
Brain distant failure (**a**), progression-free survival (**b**), and overall survival (**c**) from brain metastasis treatments (radiosurgery or surgery followed by radiosurgery), in patients with primary renal cell carcinoma.

**Table 1 cancers-15-01393-t001:** Baseline patient, tumor, and treatment characteristics.

No. pts Total (%)		
	120	100
**Patient variables**		
**Median age years** (range years)	60 (25–83)	
**Gender**		
Male	88	73.3
Female	32	26.7
**KPS**		
70	2	1.7
80	30	25
90–100	88	73.3
**Grade of RCC**		
Grade 2	47	39.2
Grade 3	57	47.5
Grade 4	16	13.3
**Stage at diagnosis**		
I–III	71	59.2
IV	49	40.8
Extracranial metastases only	37	
Brain metastases only	5	
Extracranial + brain metastases	7	
**Treatments**		
Surgery alone	81	67.5
S + Adjuvant systemic therapy	39	32.5

No. = number; pts = patients; BMs = brain metastases; KPS = Karnofsky performance scale; RCC = renal carcinoma cancer.

**Table 2 cancers-15-01393-t002:** Patient, tumor, and treatment characteristics at BM occurrence.

No. pts Total (%)		
	120	100
**Patient variables**		
**Median age years** (range years)	64 (37–84)	
KPS		
70	2	1.7
80	30	25
90–100	88	73.3
**Median IT diagnosis/BM occurrence** (range months)	28 (0–253)	
**EC metastases at BM treatments**		
No	13	10.8
Yes	107	89.2
**No EC metastatic site**		
1	30	28
2	37	34.6
3	22	20.6
4	15	14
5	2	1.9
>5	1	0.9
**RCC-specific GPA score**		
1.5–2	11	9.2
2.5–3	52	43.3
3.5–4	57	47.5
**IMDC score**		
Favorable	30	25
Intermediate	88	73.3
Poor	2	1.7
**No BMs**		
1	84	70
2	26	21.7
3	8	6.6
4	2	1.7
**BMs treatment**		
SRS 1 fraction	82	68.3
Median dose Gy (range Gy)	24 (13–25)	
HSRS 3–5 fractions	15	12.5
Median dose Gy (range Gy)	30 (21–32)	
Surgery + adjuvant SRS/HSRS	23	19.2
**Systemic treatment after RT**		
Yes	77	64.2
*ICIs*	22	
*VEGFR TKIs*	55	
No	43	35.8

No. = number; pts = patients; BMs = brain metastases; KPS = Karnofsky performance scale; IT = interval time; EC = extracranial; RCC = renal carcinoma cancer; GPA = graded prognostic assessment; IMDC = International Metastatic Database Consortium; SRS = stereotactic radiosurgery; HSRS = hypofractionated radiosurgery; RT = radiation therapy; ICIs = immune checkpoint inhibitors; VEGFR TKI = vascular endothelial growth factor receptor tyrosine kinase inhibitors.

**Table 3 cancers-15-01393-t003:** Kaplan–Meier progression-free survival (PFS) and overall survival (OS) according to subgroup analyses from BM treatments.

	No Pts	Median PFS Months (Months 95%CI)	6-Months PFS (SE)	1-Year PFS (SE)	2-Year PFS (SE)	3-Year PFS (SE)	*p* Value Univariable	HR Multivariable (95%CI)	*p* Value Multivariable
**Progression free survival (PFS)**	120	5 (4–8)	50 (±4.5)	30 (±4.1)	9.9 (±2.8)	4.1 (±2.1)			
**KPS** 70 80 90–100	2 30 88	1 (1–2) 8 (4–14) 5 (3–7)	0 56.7 (±9.0) 48.9 (±5.3)	0 36.7 (±8.8) 28.4 (±4.8)	0 0 13.9 (±3.8)	0)0 0 5.8 (±2.9)	0.0630	-	nr
**IDMC score** Favorable Intermediate Poor	30 88 2	8 (5–15) 4 (3–5) 1	68 (±9.3) 39 (±5.3) 0	40 (±9.8) 20.7 (±4.4) 0	20 (±8.0) 4.4 (±2.4) 0	8 (±5.4) 2.2 (±2.0) 0	**0.0196**	-	nr
**EC met at BMs treatments** No Yes	13 107	13 (2–24) 5 (4–7)	69.2 (±12.8) 47.7 (±4.8)	53.8 (±13.8) 27.1 (±4.3)	23.1 (±11.7) 8.3 (±2.7)	23.1 (±11.7) 3.1 (±1.7)	**0.0500**	-	nr
**Treatments** SRS/HSRS S+SRS/HSRS	97 23	5 (3–7) 17 (12–32)	41.2 (±5) 87 (±7)	20.6 (±4.1) 69.6 (±9.5)	2.0 (±1.4) 47.8 (±10.4)	1.0 (±1.0) 19.9 (±20.7)	**<0.0001**	20.0376 (0.1629–0.4919)	**<0.0001**
	**No Pts**	**Median OS months** **(months 95%CI)**	**6-months OS (SE)**	**1-year OS (SE)**	**2-year OS (SE)**	**3-year OS (SE)**	***p* value** **univariable **	**HR multivariable (95%CI)**	***p* value** **multivariable**
**Overall Survival (OS)**	120	16 (12–22)	80 (±3.6)	58.3 (±4.5)	30.9 (±4.3)	16.9 (±3.6)			
**IT** ≤12 months >12 months	34 86	22 (16–30) 14 (10–17)	82.4 (±6.5) 79.1 (±4.3)	73.5 (±7.5) 52.3 (±5.3)	42.7 (±9.0) 26.4 (±4.7)	29.9 (±8.8) 12.4 (±3.7)	**0.0251**	-	**nr**
**EC met at BMs occurrence** No Yes	13 107	26 (15–37) 16 (11–19)	100 77.6 (±4.0)	92.3 (±7.3) 54.2 (±4.8)	52.7 (±14.1) 28.3 (±4.4)	44 (±14.3) 13.8 (±3.5)	**0.0498**	-	nr
**RCC specific-GPA score** 1.5-2 2.5-3 3.5-4	11 52 57	17 (7–23) 10 (7–15) 22 (16–26)	90.9 (±8.6) 67.3 (±6.5) 89.5 (±4.0)	63.6 (±14.5) 42.3 (±6.8) 71.9 (±5.9)	9.0 (±8.6) 24.3 (±6.0) 41.4 (±6.7)	9.0 (±8.6) 8.8 (±4.1) 26.7 (±6.2)	**0.0191**	0.7170 (0.5393–0.9533)	**0.0221**
**Treatments** SRS/HSRS S+SRS/HSRS	97 23	14 (9–16) 26 (18–83)	75.3 (±4.3) 100	50.5 (±5.0) 91.3 (±5.8)	24.2 (±4.4) 59.1 (±10.6)	12.9 (±3.6) 34.5 (±10.4)	**0.0014**	0.4038 (0.2285-0.7137)	**0.0018**
**Systemic treatment after SRS/HSRS** No VEGFR TKIs ICIs	43 55 22	16 (11–19) 16 (10–22) 28 (8–29)	79.1 (±6.2) 80 (±5.3) 81.8 (±8.2)	60.5 (±7.4) 56.4 (±6.6) 59.1 (±10.5)	23.3 (±6.4) 30.1 (±6.2) 51.7 (±11.5)	9.3 (±4.4) 17.7 (±5.3) 36.9 (±12.1)	0.0540	-	nr
**ST + LT** SRS/HSRS +TKIs S+SRS/HSRS+TKIsSRS/HSRS+ICIs S+SRS/HSRS+ICIs	48 7 16 6	16 (8–22) 23 (10–83) 12 (5–23) 29 (28–29)	77.1 (±6.0) 100 75.0 (±10.8) 100	54.2 (±7.1) 71.4 (±17.1) 43.8 (±12.4) 100	28.2 (±6.6) 42.9 (±18.7) 32.8 (±13.3) 100	16.5 (±5.5) 28.6 (±17.1) 32.8 (±13.3) 50.0 (±25.0)	0.0740	-	nr

**No.** = number; pts = patients; IDMC = International Metastatic Database Consortium; KPS=Karnofsky performance scale; EC met = extracranial metastases; BMs = brain metastases; SRS = stereotactic radiosurgery; HSRS = hypofractionated radiosurgery; IT = interval time; EC = extracranial; RCC = renal carcinoma cancer; GPA = graded prognostic assessment; S=surgery; ICIs = immune checkpoint inhibitors; VEGFR TKIs = vascular endothelial growth factor receptor tyrosine kinase inhibitors; ST = systemic treatment; LT = local treatment.

## Data Availability

Data supporting the reported results are available in the dataset generated by Humanitas Research hospital and can be provided if required.

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
