# Peer review of "Multimodal Treatments for Brain Metastases from Renal Cell Carcinoma: Results of a Multicentric Retrospective Study"

_cancers, 2023, doi:10.3390/cancers15051393_

Round 1

Reviewer 1 Report

Firstly, I believe that readers would benefit from a summary that links the objectives with the study methods and their stratification.  

The manuscript will benefit from a significant change in the results representation. Especially the information that is described in the tables is difficult to interpret and unappealing for the reader.

Methodology, including statistical analysis, for the described toxicity results is missing (page 9) 

Authors are invited to complete all captions that in general are too poor in this version.

Author Response

Reviewer 1

Point 1: firstly, I believe that readers would benefit from a summary that links the objectives with the study methods and their stratification.  

Response 1: thank you for your suggestions. We integrated summary.

Point 2: The manuscript will benefit from a significant change in the results representation. Especially the information that is described in the tables is difficult to interpret and unappealing for the reader.

Response 2: done

Point 3: Methodology, including statistical analysis, for the described toxicity results is missing (page 9) 

Response 3: we integrated outcome evaluation section

Point 4: Authors are invited to complete all captions that in general are too poor in this version.

Response 4: done

Reviewer 2 Report

In this retrospective analysis, 120 pt with 176 BM from RCC were evaluated.  Patients received surgery plus postoperative HSRS or SRS, either single-fraction then hypofractionated SRS(HSRS). Local control(LC), distant brain failure(BDF), overall survival(OS), toxicities, and prognostic factors were assessed. This is an interesting analysis, although the treatment description and results are a bit difficult to follow. The manuscript is written a bit chaotic.  a large concern is the large time spam (2003-2021). as the majority of patient did also receive systemic treatment, results are difficult to interpret. For example, did ICI influence results? are more effective TKI's of influence. I could not find which patients in the different treatment groups (surg plus RT vs RT alone) received which systemic treatment.  

the aim of the analysis is not well described. line 38 "were:  assessed: were is lacking. 

how were these patients selected? were asymptomatic patients screened by MRI?  in the conclusion (line 318)  it is stated that a  multimodal treatment approach including surgery, radiosurgery and systemic therapy provided the best outcome, although this can be the result of selection of the fittest patients. 

Together it is a nice retrospective analysis, with limited consequences for clinical practice. 

Author Response

Reviewer 2

Point 1: In this retrospective analysis, 120 pt with 176 BM from RCC were evaluated.  Patients received surgery plus postoperative HSRS or SRS, either single-fraction then hypofractionated SRS(HSRS). Local control(LC), distant brain failure(BDF), overall survival(OS), toxicities, and prognostic factors were assessed. This is an interesting analysis, although the treatment description and results are a bit difficult to follow. The manuscript is written a bit chaotic.  a large concern is the large time spam (2003-2021). as the majority of patient did also receive systemic treatment, results are difficult to interpret. For example, did ICI influence results? are more effective TKI's of influence. I could not find which patients in the different treatment groups (surg plus RT vs RT alone) received which systemic treatment.  

Response 1: thank you for your comments. We tried to improve manuscript. Regarding systemic treatments, although without statistical significance, patients receiving ICI and TKis vs unreceiving ones showed better outcome as highlighted in Table 3. Relations with different kind of local treatments has been added in results section and table 3.

Point 2: the aim of the analysis is not well described. line 38 "were:  assessed: were is lacking. 

Response 2: Done

Point 3: how were these patients selected? were asymptomatic patients screened by MRI?  in the conclusion (line 318)  it is stated that a multimodal treatment approach including surgery, radiosurgery and systemic therapy provided the best outcome, although this can be the result of selection of the fittest patients. 

Response 3: in the present series the large majority of patients were symptomatic. No screening brain MRI are usually performed in this group of patients; notwithstanding in extracranial metastatic patients total body CT scan allow us to discover small BMs in asymptomatic patients in which MRI has been performed consequentially. We agree with you that the better outcome obtained could be the result of a selection of fittest patients. However, majority of patients showed KPS 80-100 and the selection criteria for surgical resection was mainly related to BMs characteristics and extra cranial disease status